# Predictors of frequency of CF care in the US Cystic Fibrosis Foundation Patient Registry

Alexandra C. Hinton [1,2]*, Edmund H. Sears[3], Jonathan B. Zuckerman[3], Sara Lopez-Pintado[2]

1 Center for Interdisciplinary Population and Health Research, MaineHealth Institute for Research, Scarborough, Maine, United States of America, 2 Bouve College of Health Sciences, Northeastern University, Boston, Massachusetts, United States of America, 3 Pulmonary and Critical Care Medicine, MaineHealth Maine Medical Center, Portland, Maine, United States of America

* alexandra.hinton@mainehealth.org

**Data Availability Statement:** Data cannot be shared publicly because of patient privacy concerns. Data are available from the USCFFPR Comparative Effectiveness Research Committee (contact via email: datarequests@cff.org) for

## Abstract

### Introduction

Prolonged gaps in care of >12-months are frequent among people with cystic fibrosis (pwCF) and are associated with reduced lung function. Comprehensive analysis of patient-level predictors of visit frequency is needed to optimize protocols for stable pwCF and identify subgroups at high risk of gaps and poor outcomes, promoting equitable treatment for all pwCF.

### Objective

To determine sociodemographic and disease-related factors predictive of visit frequency in pwCF and to assess how these effects vary across the lifespan.

### Methods

We conducted an observational cohort study using data from 2004–2016 for pwCF aged 6–60 years in the US Cystic Fibrosis Foundation Patient Registry. We modeled the relationship between patient-level characteristics and between-visit interval (BVI) using multivariable longitudinal semiparametric regression. BVI was defined as the number of days from the index encounter to the previously recorded visit.

### Results

The study included 28,588 pwCF with 859,568 encounters. Overall, 55% of visits occurred within 90 days of the prior visit, adhering to national guidelines. On average, adults without common CF-complications attended clinic approximately every 4 months, with a BVI $\geq$ 110 days from age 23–56. Males attended clinic less frequently than females (9.8% longer BVI; 95% CI 9.1%, 10.5%; p<0.001), as did non-white individuals (3.6% longer BVI than whites; 95% CI 2.2%, 5.0%; p<0.001), with the greatest differences seen in young adults. Those with public and private insurance largely adhered to current guidelines (maximum average BVI of 90 and 95 days, respectively). In contrast, uninsured individuals over age 25 had a mean BVI $\geq$ 30 days longer than the insured.

researchers who meet the criteria for access to confidential data.

**Funding:** AM and JZ received funding from the Cystic Fibrosis Foundation (https://www.cff.org/) through the StatNet grant 003847Y7122. SLP received funding from the National Science Foundation, grant NSF DMS-2113696. The funders did not play any role in the study design, data collection and analysis, decision to publish, or preparation of the manuscript.

**Competing interests:** The authors have declared that no competing interests exist.

## Conclusions

Frequent visits in those with CF-complications likely reflects higher need, while less frequent visits in male, non-white, and uninsured individuals may reflect patient-preference or structural barriers to care. Risk factors for gaps in care should inform changes to CF care recommendations going forward.

## Introduction

Understanding variability in clinic visit frequency among people with cystic fibrosis (pwCF) is crucial for optimizing long-term care management. This knowledge is essential for developing targeted interventions to support patients at high risk of missed visits, addressing socioeconomic and geographical barriers to promote equitable access to care. Recent evidence underscores the importance of this issue. Prolonged gaps in care exceeding 12 months are common among pwCF, and these interruptions are associated with reduced lung function, independent of other factors known to be associated with pulmonary compromise [1]. Despite its importance, there is currently a notable scarcity of literature on this topic.

Additionally, insights into missed visits can inform refinement of clinical guidelines and health policies, leading to realistic and achievable care recommendations. Current guidelines published by the Cystic Fibrosis Foundation (CFF) recommend that pwCF annually attend at least four CF Center clinic appointments at approximately 90-day intervals [2]. These recommendations are based on expert opinion and retrospective data from a study completed nearly 20 years ago [3, 4]. To meet these guidelines, many CF centers have launched resource- and time-intensive quality improvement initiatives to increase outpatient visit frequency [5, 6]. Research examining the predictors of clinic visit frequency could improve the cost-effectiveness of these interventions.

Since 2004, the US Cystic Fibrosis Foundation Patient Registry (USCFFPR) has tracked encounter-based data for pwCF to enhance healthcare delivery and facilitate quality improvement and research [7]. This registry captures both outpatient and inpatient contacts with the US CF healthcare system, along with pulmonary function testing (PFTs) and a detailed body of clinical and demographic information. This comprehensive database is an excellent resource for investigating characteristics associated with variations in between-visit intervals (BVI) among pwCF.

A comprehensive analysis of patient-level factors influencing visit frequency will inform future care recommendations and promote equity in CF care. This study describes predictors and age-related patterns of visit frequency in pwCF utilizing data from the USCFFPR. Preliminary findings have been presented as an abstract [8].

## Materials and methods

### Study population

This study was approved by the USCFFPR committee and the Maine Medical Center Institutional Review Board. The requirement for informed consent was waived because the study used de-identified registry data. We obtained USCFFPR data on July 5, 2018 for encounters which spanned January 1, 2004 through December 31, 2016. All participants enrolled in the USCFFPR had given written informed consent (or in the case of minors, assent along with parental or guardian consent) to their CF care center. Data are entered prospectively into a

```
┌─────────────────────────────────────────────────────┐
│              Encounters in the USCFFPR                │
│                     2004-2016                         │
│         1,636,912 encounters (28,861 people)          │
└─────────────────────────────────────────────────────┘
                           │
                           ▼
┌─────────────────────────────────────────────────────┐
│ Exclusions:                                           │
│ • Age <6 or age > 60: 191,879 encounters              │
│ • Pre (1 year) and post lung transplant: 53,478       │
│   encounters                                          │
│ • Missing PFT or BMI:                                 │
│ • 302 total encounters                                │
│        - Unable to impute PFT (47 encounters; 8       │
│          people)                                      │
│        - Unable to impute BMI (295 encounters; 12     │
│          people)                                      │
│ • Did not have at least 2 total encounters:           │
│   41 encounters; 41 patients                          │
│ • Did not have at least 30 days between encounters:    │
│   531,644 encounters                                  │
└─────────────────────────────────────────────────────┘
                           │
                           ▼
┌─────────────────────────────────────────────────────┐
│                   Study Cohort                        │
│        859,568 encounters (28,588 patients)           │
└─────────────────────────────────────────────────────┘
```

**Fig 1. Flow chart of the study cohort, derived from people with cystic fibrosis in the United States Cystic Fibrosis Foundation Patient Registry between 2004 and 2016.** The figure includes a description of patient-level and encounter-level exclusions. BMI = body mass index; CF = cystic fibrosis; USCFFPR = United States Cystic Fibrosis Foundation Patient Registry; PFT = pulmonary function test.

secure database by care center staff. Subjects entered the study cohort from January 1, 2004 to December 31, 2014 and were included until one year prior to lung transplantation (when care is often performed at transplant centers), death, loss to follow-up, or the end of the study period on December 31, 2016. Authors had access to potentially identifying information only if critical to the research project, such as zip code, encounter date, and age at encounter. Missing data were handled using the last observation carried forward (LOCF) and, subsequently, next observation carried backward (NOCB) imputation techniques. After imputation, we excluded pwCF missing BMI and PFT measurements. Those age > 60 years were excluded due to increased frequency of rare genotypes and delayed diagnosis. We restricted our dataset to individuals with at least two visits recorded, as calculating the time interval between visits requires a minimum of two encounters (Fig 1). We followed Strengthening the Reporting of Observational Studies in Epidemiology (STROBE) guidelines.

## Outcome and covariates

The outcome, between-visit interval (BVI), was calculated for each encounter as days from the prior encounter in the USCFFPR. Visits with BVI <30 days were excluded, as USCFFPR data did not specify inpatient or outpatient settings, and such short intervals typically indicate acute issue management rather than routine longitudinal care. We log-transformed the outcome to address skewed data distributions and mitigate the influence of outliers.

Our selection of potential predictors was informed by prior research showing substantial impact on clinic attendance in pwCF and other chronic conditions [9, 10] and the authors' clinical observations. Demographic predictors were age, sex, race, income, education, rurality as defined by rural-urban community area categorization A [11], and insurance type. To simplify the investigation of a three-way interaction between age, race, and insurance type, we created a consolidated variable merging race and insurance type. Disease-related factors included cystic fibrosis transmembrane conductance regulator (*CFTR*) genotype, underweight body mass index (BMI), CF-related diabetes (CFRD), pulmonary impairment (based on percent predicted forced expiratory volume in one second; FEV1PP), chronic lung infections, and a composite variable capturing common CF complications (severe lung impairment [FEV1PP $\leq$ 40], underweight BMI, CFRD, and chronic infection). Age, sex, race, and genotype were static predictors. All other factors were treated as time-varying, and we used values from the preceding visit to maintain temporal precedence, supporting potential causal inferences. Last observation carried forward (LOCF) and next observation carried backward (NOCB) was applied to BMI, FEV1PP, family income, education, and rurality. Where LOCF and NOCB were not possible due to a complete lack of information across the study period for an individual, we included these cases using an unknown/missing category. This approach allowed us to retain valuable data from all participants, even those with limited information. S1 Table provides variable definitions and rules for missing data imputation.

## Statistical analysis

We summarized cohort characteristics using n (%) for categorical variables. We constructed separate multivariable models to test the association between BVI and age, sex, race, insurance type, rurality, *CFTR* genotype, underweight BMI, CFRD, pulmonary impairment, chronic lung infections, and CF-related complications at the visit prior to the index encounter (**Models 1–15**). Confounding was assessed separately for each predictor-outcome pairing using directed acyclic graphs (see example in S1 Fig). To account for the progressive nature and individual variability of CF, we employed longitudinal mixed effects regression methods. These models accounted for time-varying covariates and random effects at the person and center levels. Since the outcome was log-transformed, we reported the percent difference in the BVI compared to the referent group, calculated as $100(e^{\beta} - 1)\%$. To assess changes in these relationships across the lifespan, we added an interaction term between the predictor of interest and age, where age was modeled using natural cubic splines (knots positioned at the quantiles) to allow for nonlinear behavior (**Models 1a – 15a**). As age was modeled with splines, we tested interaction effects using type III analysis of variance (ANOVA). Estimated BVI for the strata of the predictor of interest was plotted against age by conditioning on a weighted average of other covariates [12]. We conducted ad hoc investigations into the interaction between race and insurance type on BVI as well as the number of concurrent chronic infections on BVI (**Models 6 and 14**). We assessed changes in these relationships across the lifespan (**Models 6a and 14a**). To obtain pairwise comparisons of factor levels, we used estimated marginal means. These means were computed from the fitted model, adjusting for covariates and the model structure and were not adjusted for multiple comparisons.

Five sensitivity analyses were conducted to evaluate the effects of analytic decisions. We separately modeled the effect of predictors 1) when excluding any encounters with data missing prior to imputation for any predictors of interest (complete case analysis), 2) when including encounters with BVI < 30 days (an exclusion criterion in our primary analysis), 3) in a subset of encounters where pwCF were experiencing no CF-related complications, 4) in

pediatric and adult populations, and 5) when dichotomizing education (households with or without one member with a college degree).

In an ad hoc analysis, we examined pwCF with at least one G551D *CFTR* mutation who became eligible for ivacaftor in 2012. While we did not have data on ivacaftor use, we did have mutation information, and uptake of ivacaftor for patients with G551D was rapid [13]. To ensure sufficient data, we constructed a histogram of individual pwCF with G551D contributing to each age. We then modeled BVI as a function of age (modeled with splines) and time-period (a binary indicator for pre- and post-2012 periods). This approach allowed us to assess the impact of ivacaftor availability on care utilization patterns while accounting for age-related trends. We included an interaction term between age and time-period in the model to plot this association.

We performed all analyses in R version v4.2.1 (R Core Team 2021). Packages used included gtsummary for tables, nlme and splines for data analysis, and effects, emmeans, and ggplot2 for model predictions and visualizations. Model specification, example code, and descriptions of sensitivity analyses can be found in the supplemental materials (S1 File).

## Results

### Study population

Inclusion criteria were met for 28,588 pwCF with 859,568 encounters in the USCFFPR (Fig 1). Demographics, shown in Table 1, are reflective of the overall US CF population. A comparison of the study cohort to the 67 excluded individuals showed that those not included were more likely to be very young and missing additional data (S2 Table).

### Prevalence of prolonged periods without care

Approximately half (55%) of the encounters in this study occurred within the recommended 3-month window, about a quarter (28%) between 3–4 months, and 8% ≥ 6 months (S3 Table). Most of the population experienced at least one 6-month BVI (n = 18,047; 63%). More than a quarter were not seen for at least one yearlong span during the study (n = 8,404; 29%), and 16% had a BVI of ≥ 18 months (n = 4,584). Pediatric pwCF were less likely than adults to have prolonged BVI, but proportions of those going 6, 12, and 18 months without an encounter were still considerable (42%, 12%, and 5% versus 67%, 34% and 20%, respectively; S4 Table). The percentage of pediatric pwCF with BVI ≥ 6 months increased with age; 16% of those aged 6–8, 24% of those aged 12–14, and 28% of those aged 16–18 (S5 Table).

### Predictors of between visit interval

Multivariable models (Table 2, Fig 2) show the overall effect of each covariate on BVI adjusted for confounding (confounding adjustment sets in S6 Table and Table 2). Figs 3–5 display adjusted differences in BVI across the lifespan for predictor strata. Statistically significant interaction was found between age and all predictors evaluated (all p<0.02).

**Changes in between visit interval over the lifespan.** Importantly, BVI increased significantly in early adulthood, with an inflection around age 18. Children were typically seen within the 90-day timeframe recommended by CFF guidelines. On average, BVI began to increase around age 15, crossing the 90-day threshold at 22 and remaining above that level through the early 30's (Panel a in Fig 3). This overarching pattern of BVI inflection in late adolescence-early adulthood was observed across multiple subgroups, though it did not always result in crossing the 90-day threshold (Figs 3–5).

**Table 1. Characteristics of people with cystic fibrosis aged 6–60 years old in the United States Cystic Fibrosis Foundation Patient Registry from 2004–2016 included in the study.**

| Sociodemographic factors | Overall N = 28,588[1] |
|---|---|
| **Birth Cohort** | |
| Before 1981 | 5,629 (20%) |
| 1981–1988 | 5,558 (19%) |
| 1989–1994 | 5,376 (19%) |
| 1995 and after | 12,025 (42%) |
| **Sex** | |
| Male | 14,845 (52%) |
| **Race/Ethnicity** | |
| White race | 26,926 (94%) |
| Hispanic | 2,011 (7.0%) |
| Black or African American | 1,256 (4.4%) |
| American Indian or Alaska Native | 195 (0.7%) |
| Asian | 156 (0.5%) |
| Native Hawaiian or Other Pacific Islander | 16 (<0.1%) |
| Other race | 426 (1.5%) |
| **Rurality[2]** | |
| Urban | 21,802 (76%) |
| Large rural | 3,210 (11%) |
| Small rural | 1,819 (6.4%) |
| Isolated | 1,524 (5.3%) |
| Missing/Unknown | 233 (0.8%) |
| **Insurance Coverage*** | |
| Public Insurance | 18,370 (64%) |
| Private Insurance | 23,141 (81%) |
| Other Insurance | 3,500 (12%) |
| Unknown Insurance | 494 (1.7%) |
| No Insurance | 1,572 (5.5%) |
| **Highest Education in Family[3]** | |
| Less than High School | 508 (1.8%) |
| High School diploma or equivalent | 3,694 (13%) |
| Some College | 4,769 (17%) |
| College Graduate | 12,362 (43%) |
| Masters/Doctoral level degree | 5,701 (20%) |
| Unknown | 1,554 (5.4%) |
| **Maximum Family Income[4]** | |
| <$40,000 | 4,777 (17%) |
| $40,000 to $90,000 | 5,005 (18%) |
| >$90,000 | 5,679 (20%) |
| Unknown | 13,127 (46%) |
| *Disease-related factors* | |
| **CFTR Genotype** | |
| F508del heterozygote | 10,920 (38%) |
| F508del homozygote | 12,994 (45%) |
| Other/Unknown | 4,674 (16%) |
| **Pulmonary Impairment[5]** | |
| Mild (FEV1PP ≥70%) | 10,645 (37%) |

*(Continued)*

**Table 1.** (Continued)

| Sociodemographic factors | Overall N = 28,588[1] |
|---|---|
| Moderate (FEV1PP 41–69%) | 9,809 (34%) |
| Severe (FEV1PP ≤ 40%) | 8,134 (28%) |
| **Underweight**[*6] | 9,505 (33%) |
| **Chronic Infection**[*] | |
| *P. aeruginosa* | 18,363 (64%) |
| MRSA | 9,590 (34%) |
| *Burkholderia* spp. | 1,606 (5.6%) |
| **Number of Chronic Infections**[*] | |
| No chronic infections | 7,189 (25%) |
| Single pathogen | 13,648 (48%) |
| Two | 7,342 (26%) |
| Three | 409 (1.4%) |
| **CF-Related Diabetes**[*] | 9,410 (33%) |
| **CF-related Complication**[*7] | 23,343 (82%) |
| **Time in study (years)** | |
| Median (IQR) | 9.6 (5.4, 12.5) |
| Mean (SD) | 8.7 (3.8) |
| Range | 0.1, 13.0 |
| **Patient Died During Study Period** | 2,516 (8.8%) |

[1] n (%), unless otherwise specified

[2] Most rural residential stratum during the study period

[3] Highest level of education among the individual, their parents, and/or spouse during the study period

[4] Highest reported income during the study period

[5] Lowest lung function stratum during study period

[6] BMI< 18.5 for adults or BMI Percentile < 5% for children

[7] Includes severe lung impairment (FEV1PP ≤ 40), underweight BMI, CF-Related Diabetes, and chronic infection

*Ever experienced during the study period. Total within the category may therefore exceed 100%.

MRSA = Methicillin-resistant *Staphylococcus aureus*, FEV1PP = Forced expiratory volume in one second,

IQR = Interquartile range, SD = Standard deviation

**Associations between sociodemographic factors and between visit intervals.** *Sex.* On average, males attended clinic less frequently than females (9.8% longer BVI; 95% CI 9.1%, 10.5%; p<0.001). This difference became statistically significant at age 8 and persisted through adulthood, peaking between ages 26 and 32 (average difference <7% or 6 days in childhood; 17–18% or 15 days in early adulthood, Panel b in Fig 3).

*Race & ethnicity.* Non-white pwCF (Hispanic, Black, Asian, Native American, Pacific Islander, and other race) attended CF clinic with similar frequency to whites until their late teens. Modest differences in BVI of 3–7 days were seen in non-white and white individuals between ages 16–32 (overall average difference of 3.6%; 95% CI 2.2%, 5.0%; p <0.001; Panel c in Fig 3).

*Rurality.* Rurality had a small overall effect on age-related BVI patterns. Those living in non-urban areas attended clinic slightly less frequently than urban dwellers (Panel d in Fig 3).

*Insurance type.* Insurance type was an important predictor of BVI (Panel a in Fig 4). Publicly and privately insured pwCF largely adhered to the recommended 90-day BVI guidelines, with a maximum average BVI of 90 and 95 days, respectively. Those with public and private

**Table 2. Estimates of the total effect of predictors of interest on the interval between CF clinic visits among those aged 6–60 years old in the United States Cystic Fibrosis Foundation Patient Registry from 2004–2016.** These estimates are from multivariable models (**Models 1–15**), each with individualized confounder adjustment.

| | % Δ in BVI [1] | 95% CI | e[β] | 95% CI | P value |
|---|---|---|---|---|---|
| *Sociodemographic factors* | | | | | |
| **Age at Encounter (decades, linear) [2],[3]** | -0.6 | -0.8, -0.3 | 0.99 | 0.99, 1.00 | <0.001 |
| **Sex [3]** | | | | | |
| Male (female referent) | 9.8 | 9.1, 10.5 | 1.10 | 1.09, 1.11 | <0.001 |
| **Race/Ethnicity [3]** | | | | | |
| White and non-Hispanic (referent) | | | | | |
| Non-white or Hispanic | 3.6 | 2.2, 5.0 | 1.04 | 1.02, 1.05 | <0.001 |
| **Rurality [3]** | | | | | |
| Urban (referent) | | | | | |
| Large rural | 1.7 | 0.9, 2.6 | 1.02 | 1.01, 1.03 | <0.001 |
| Small rural | 4.0 | 2.8, 5.1 | 1.04 | 1.03, 1.05 | <0.001 |
| Isolated | 1.2 | -0.1, 2.5 | 1.01 | 1.00, 1.02 | 0.063 |
| Unknown | 8.4 | 6.4, 10.4 | 1.08 | 1.06, 1.10 | <0.001 |
| **Insurance Coverage [4]** | | | | | |
| Private (referent) | | | | | |
| Public Insurance | -3.4 | -3.8, -3.0 | 0.97 | 0.96, 0.97 | <0.001 |
| Other Insurance | 3.5 | 1.8, 5.1 | 1.03 | 1.02, 1.05 | <0.001 |
| No insurance | 19.4 | 17.6, 21.2 | 1.19 | 1.18, 1.21 | <0.001 |
| Unknown Insurance Status | 13.0 | 11.8, 14.2 | 1.13 | 1.12, 1.14 | <0.001 |
| **Insurance/Race [5]** | | | | | |
| Private, White versus Non-White | -3.8 | -5.3, -2.2 | 0.96 | 0.95, 0.98 | <0.001 |
| Public, White versus Non-White | -4.8 | -6.2, -3.3 | 0.95 | 0.94, 0.97 | <0.001 |
| Other, White versus Non-White | -3.3 | -6.1, -0.4 | 0.97 | 0.94, 1.00 | 0.028 |
| **Education [6]** | | | | | |
| Less than high school education (referent) | | | | | |
| High School diploma or equivalent | 0.3 | -2.6, 3.2 | 1.00 | 0.97, 1.03 | 0.9 |
| Some College | -0.4 | -3.3, 2.5 | 1.00 | 0.97, 1.03 | 0.8 |
| College Graduate | 1.5 | -1.3, 4.4 | 1.02 | 0.99, 1.04 | 0.3 |
| Masters/Doctoral level degree | 1.9 | -0.9, 4.9 | 1.02 | 0.99, 1.05 | 0.2 |
| Missing | 3.3 | -0.1, 6.7 | 1.03 | 1.00, 1.07 | 0.057 |
| **Income [7]** | | | | | |
| <$40,000 (referent) | | | | | |
| $40,000 to $90,000 | 0.5 | -0.2, 1.2 | 1.01 | 1.00, 1.01 | 0.14 |
| >$90,000 | 1.1 | 0.3, 1.9 | 1.01 | 1.00, 1.02 | 0.010 |
| Missing | 3.1 | 2.2, 4.0 | 1.03 | 1.02, 1.04 | <0.001 |
| *Disease-related factors* | | | | | |
| **CFTR Genotype [8]** | | | | | |
| F508del Heterozygote (referent) | | | | | |
| F508del Homozygote | -4.6 | -5.2, -3.9 | 0.95 | 0.95, 0.96 | <0.001 |
| Other or Unknown Mutation | 4.5 | 3.5, 5.6 | 1.05 | 1.04, 1.06 | <0.001 |
| **Pulmonary Impairment [9]** | | | | | |
| Mild (FEV1PP ≥ 70%; referent) | | | | | |
| Moderate (FEV1PP 41–69%) | -14.6 | -14.9, -14.3 | 0.85 | 0.85, 0.86 | <0.001 |
| Severe (FEV1PP ≤ 40%) | -27.6 | -28.0, -27.2 | 0.72 | 0.72, 0.73 | <0.001 |
| **Underweight [10]** | -9.3 | -9.8, -8.8 | 0.91 | 0.90, 0.91 | <0.001 |
| **CF-related Diabetes [11]** | -16.2 | -16.6, -15.8 | 0.84 | 0.83, 0.84 | <0.001 |

(*Continued*)

**Table 2.** (Continued)

| | % Δ in BVI [1] | 95% CI | $e^{\beta}$ | 95% CI | *P* value |
|---|---|---|---|---|---|
| **Chronic Infection [12]** | | | | | |
| *P. aeruginosa* | -10.5 | -10.9, -10.1 | 0.90 | 0.89, 0.90 | <0.001 |
| MRSA | -10.4 | -10.8, -10.0 | 0.90 | 0.89, 0.90 | <0.001 |
| *Burkholderia* spp. | -10.2 | -11.1, -9.3 | 0.90 | 0.89, 0.91 | <0.001 |
| **Number of Chronic Infections [12]** | | | | | |
| No chronic infections (referent) | | | | | |
| Single pathogen | -11.1 | -11.5, -10.7 | 0.89 | 0.88, 0.89 | <0.001 |
| Two | -20.1 | -20.5, -19.6 | 0.80 | 0.79, 0.80 | <0.001 |
| Three | -24.5 | -25.9, -23.0 | 0.76 | 0.74, 0.77 | <0.001 |
| **Prior CF-related Complications [13]** | | | | | |
| Complications (referent) | | | | | |
| No complications | 16.0 | 15.4, 16.5 | 1.16 | 1.15, 1.16 | <0.001 |

[1] Adjusted percent difference in between visit interval = $100(\exp(\beta) - 1)\%$

[2] Per 10 year increase in age, linear model without use of splines

[3] No confounding adjustment

[4] Insurance: Adjusted for age, non-white, education, income

[5] Insurance/Race: Adjusted for age, education, income

[6] Education: Adjusted for age, non-white

[7] Income: Adjusted for sex, non-white, education, rurality

[8] Genotype: Adjusted for non-white

[9] Pulmonary impairment: Adjusted for age, sex, genotype, underweight, chronic infections

[10] Underweight: Adjusted for age, sex, income, genotype, CFRD, chronic infections

[11] CF-related diabetes: Adjusted for age, sex, genotype

[12] Chronic infections: Adjusted for age, sex, CFRD

[13] Prior CF-related Complications: Adjusted for age, sex, insurance, genotype

BVI = between-visit interval, MRSA = Methicillin-resistant *Staphylococcus aureus*, FEV1PP = Forced expiratory volume in one second, CFRD = cystic fibrosis-related diabetes

insurance had similar patterns of BVI in childhood; however, BVI began to increase around age 19 in those with private insurance and remained 7 days longer than those with public plans throughout adulthood. Compared to insured pwCF, uninsured individuals over 25 years of age had a mean BVI that was ≥ 30 days longer. For example, model-predicted BVI for an average 40-year-old uninsured pwCF is 125 days, less than three encounters annually instead of the recommended four visits. An ad hoc analysis showed clear interaction between race and insurance with non-white adults having lengthier BVI compared to white adults across insurance types, especially among those without clearly identified insurance plans (other, unknown and no insurance; Panel b in Fig 4).

*Education*. We did not find evidence of an association between family education level (pwCF, spouse, and parents) and BVI (Panel c in Fig 4).

**Income.** BVI increased with higher family income; however, 46% of the study cohort lacked income data (Panel d in Fig 4).

**Disease-related factors and between visit interval.** *CFTR genotype*. Individuals homozygous for the F508del mutation attended clinic more frequently over the lifespan than heterozygotes (-4.6% difference, 95% CI: -5.2, -3.9, p<0.001). Those with unknown or other CF

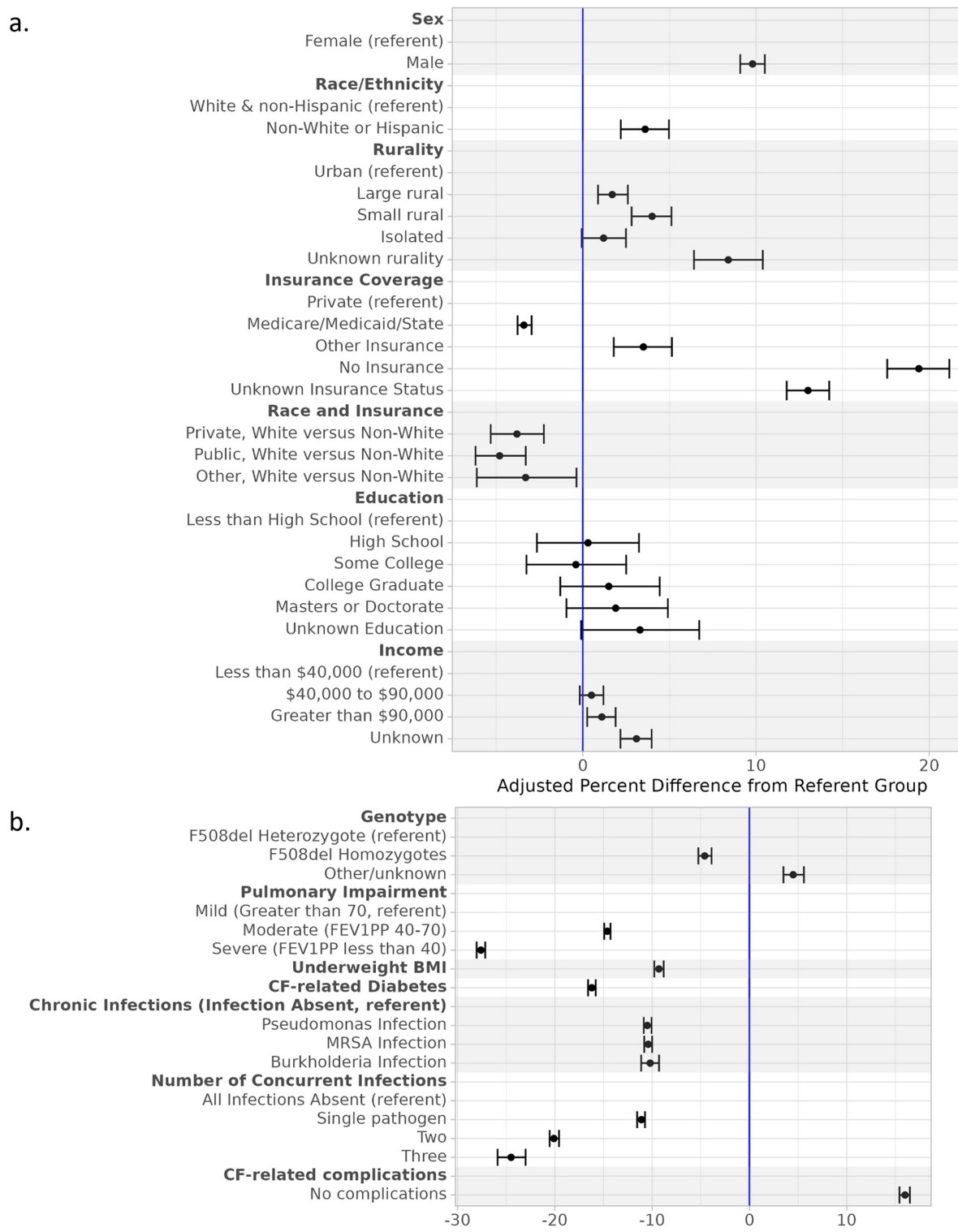

**Fig 2.** Graphical representation of multivariable model results for **a.)** sociodemographic factors and **b.)** disease-related factors and their association with the interval between visits (BVI). Results are adjusted for confounding (individually assessed) and include random effects for subject and CF center (**Models 1–15**). All models include 28,588 pwCF with 859,568 encounters.

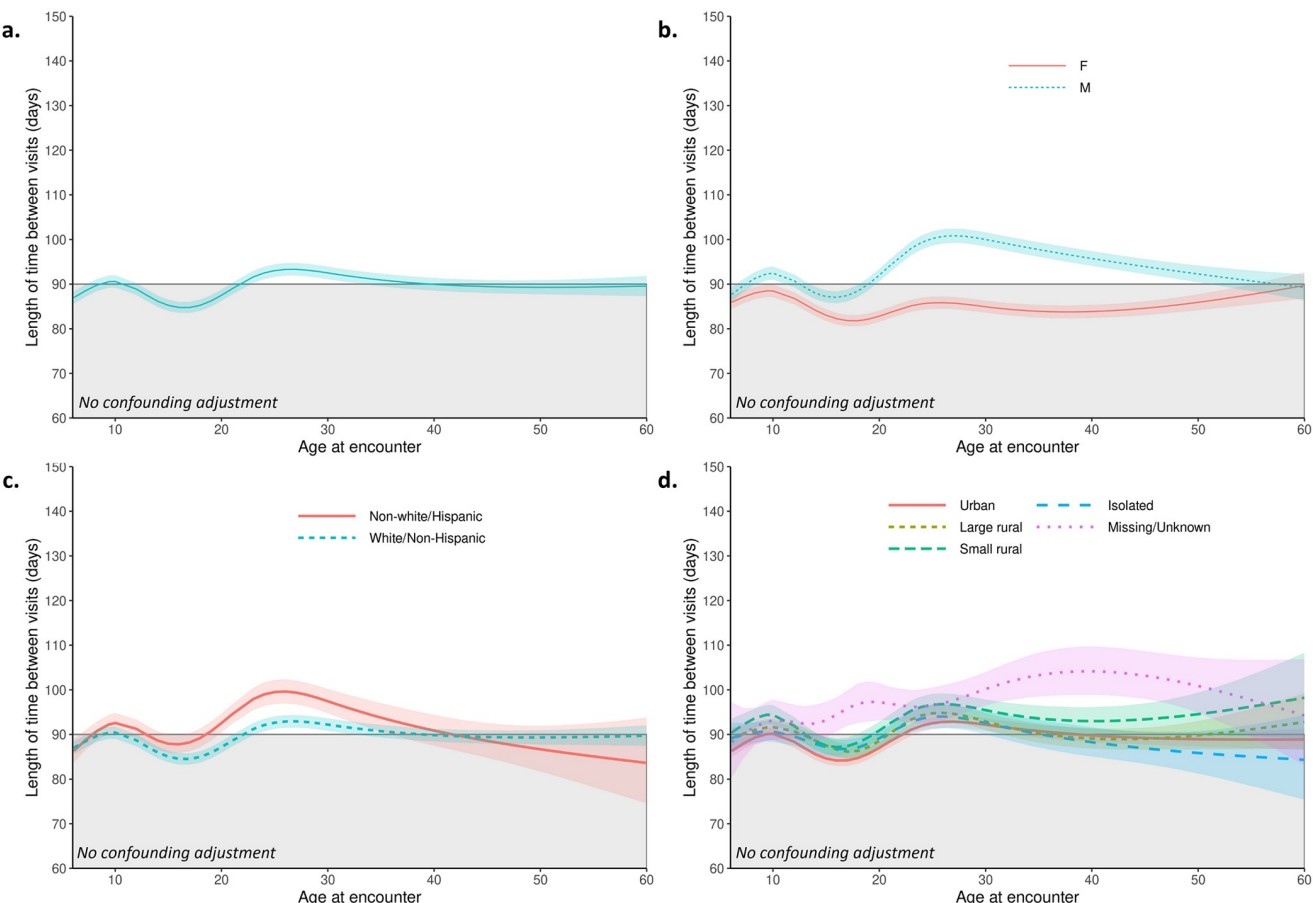

**Fig 3. Associations between sociodemographic factors and changes in between-visit interval (BVI) over the lifespan.** Color shaded areas indicate 95% confidence bounds. The gray shaded areas indicate the current CF Foundation-recommended between-visit interval of 90 days for reference. Estimated between-visit values are given for a typical person (i.e., at average values for covariates). **a.)** Estimated between-visit interval curve across age modeled with splines (**Model 1a**); **b.)** by sex (**Model 2a**); **c.)** by race (**Model 3a**); **d.)** by rurality (**Model 4a**).

genotypes had the longest BVI, 9.6% longer BVI (95% CI 1.08%, 1.11%; p<0.001) than F508del homozygotes (Panel b in Fig 2, Panel a in Fig 5).

*Disease-related complications.* Patients who were underweight (-9.3%; 95%CI –9.8, -8.8; p<0.001); had CF-related diabetes (-16.2%; 95% CI -16.6%, -15.8%; p<0.001); or were chronically infected with *Pseudomonas aeruginosa* (-10.5%; 95% CI –10.9%, -10.1%; p<0.001), MRSA (-10.4%; 95% CI –10.8%, -10.0%; p<0.001), or *Burkholderia* spp. (-10.2%; 95% CI – 11.1%, -9.3%; p<0.001) attended clinic more frequently than those without each of these disease-related complications. An ad hoc analysis showed that BVI decreased with increasing number of concurrent infections (Table 2, Panel e in Fig 5). Compared to infection-free pwCF, BVI was reduced by 11.1% (95% CI: 10.7% to 11.5%) with one pathogen, 20.1% (19.6% to 20.5%) with two, and 24.5% (23.0% to 25.9%) with three; all p<0.001. Similarly, those with moderate (-14.6%; 95% CI –14.9%, -14.3%; p<0.001) and severe (-27.6%; 95% CI -28.0%, -27.2%; p<0.001) airflow impairment attended more frequently than those with mild impairment. Compared to adults with common CF-complications (severe lung impairment, underweight BMI, CF-related diabetes, or chronically infected with *Pseudomonas aeruginosa*, MRSA, or *Burkholderia* spp), those without complications went 16% longer between visits

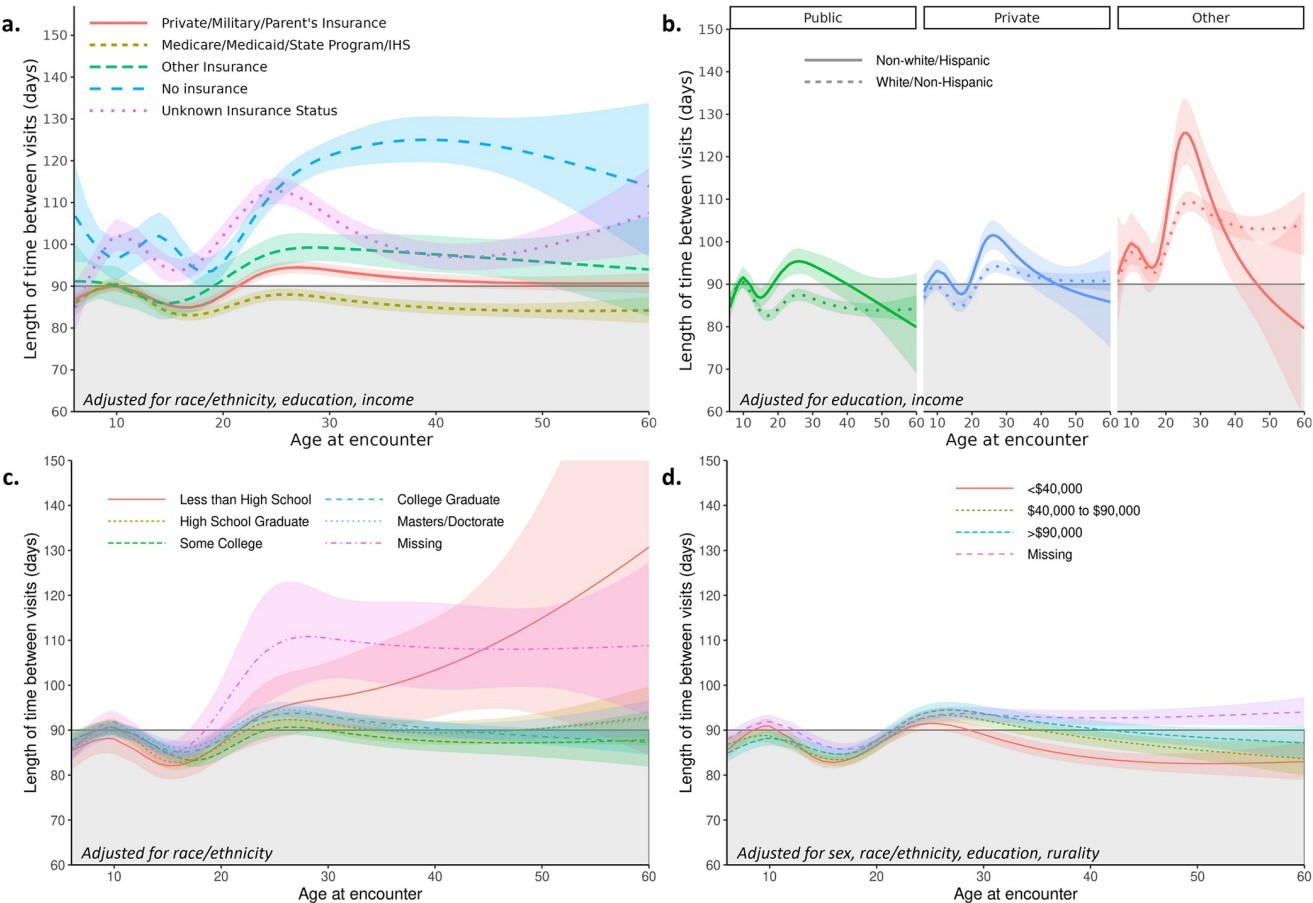

**Fig 4. Associations between sociodemographic factors and changes in between-visit interval (BVI) over the lifespan.** Color shaded areas indicate 95% confidence bounds. The gray shaded areas indicate the current CF Foundation-recommended between-visit interval of 90 days for reference. Estimated between-visit values are given for a typical person (i.e., at average values for covariates). Estimated between-visit interval curve across age modeled with splines **a.)** by insurance type (**Model 5a**); **b.)** by insurance type and race (**Model 6a**); **c.)** by education (**Model 7a**); and **d.)** by income (**Model 8a**).

(95% CI 15.4%, 16.5%; p<0.001). This translates to clinic attendance nearly every 4 months on average rather than the recommended 3-month interval (BVI 110–120 days from ages 23–56).

## Sensitivity analyses

A complete case analysis, excluding 81% of encounters and 48% of individuals, yielded similar results for disease-related characteristics but showed longer BVI for older age, advanced education, and higher income (S7 Table). This discrepancy supports our decision to impute values, as complete cases likely overrepresented individuals with higher socioeconomic status and better healthcare access, potentially biasing results. To ensure that our decision to exclude visits separated by ≤30 days had the intended effect (i.e. focusing on routine rather than sick visits), the multivariable models were run without this restriction. Findings were similar in associative direction and, as expected, adverse disease-related characteristics showed stronger associations with more frequent visits, while other characteristics showed weaker associations (S7 Table).

Analyses conducted in a subset of encounters where pwCF had no CF-related complications (S7 Table) revealed similar predictors of BVI; however, age was significantly associated

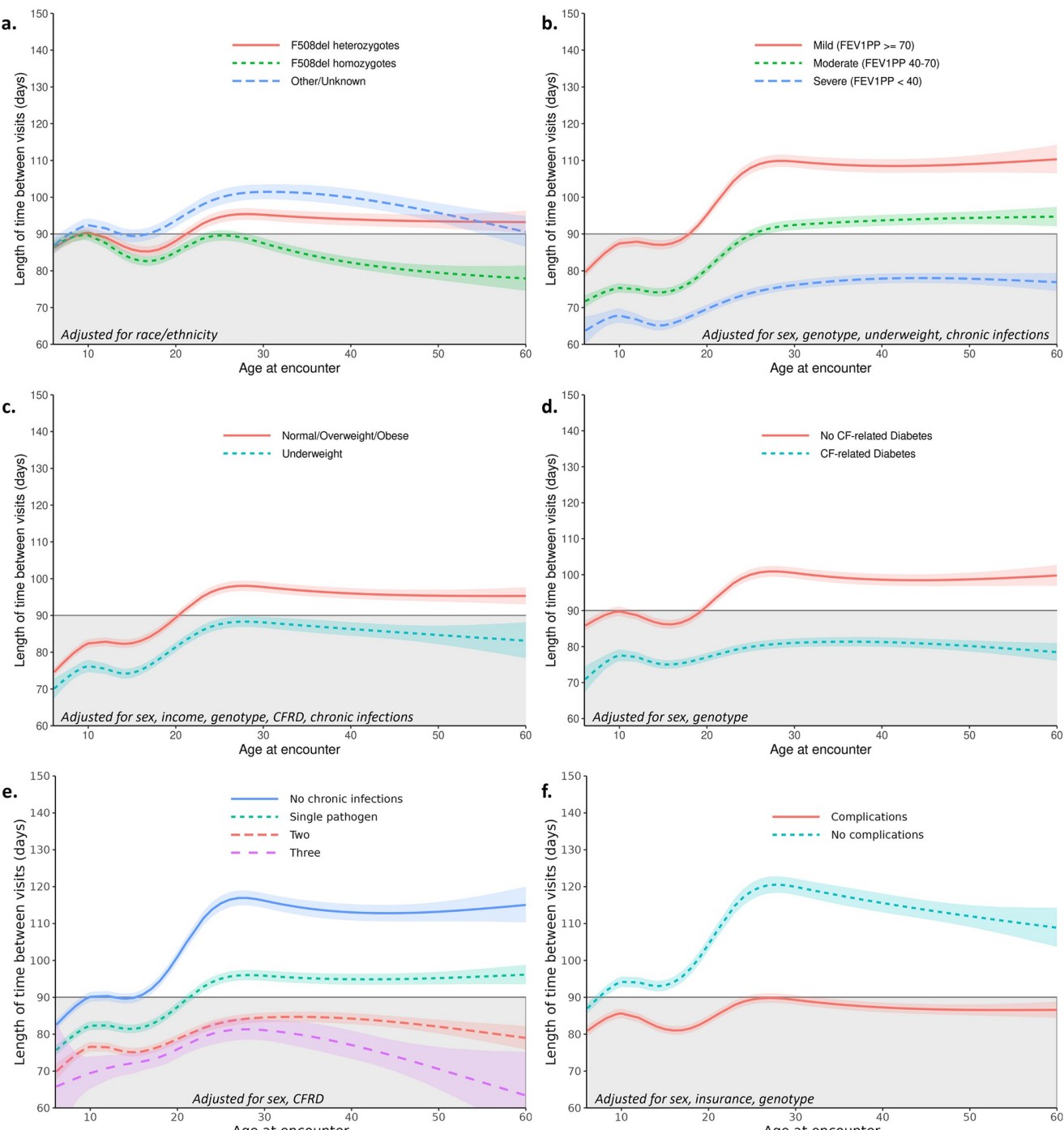

**Fig 5. Associations between disease-related factors and changes in between-visit interval over the lifespan.** The tinted areas indicate 95% confidence bounds. The gray shaded areas indicate the current CF Foundation-recommended between-visit interval of 90 days for reference. Estimated between-visit values are given for a typical person (i.e., at average values for covariates). Estimated between-visit interval curves **a.)** by genotype (**Model 9a**); **b.)** by lung impairment (**Model 10a**); **c.)** by weight category (**Model 11a**); **d.)** by CF-related diabetes status (**Model 12a**); **e.)** by number of chronic infections (**Model 14a**); and **f.)** by the presence or absence of any medical complications (**Model 15a**).

with longer BVI, as were families with college-level education and advanced degrees when compared to those with less than high school-level education. Analyses within pediatric and adult subgroups were similar, with notable differences in the effects of sex and insurance on BVI (S8 Table), the discrepancies being more pronounced in both categories in adults. Dichotomized family education levels (with/without college degree) showed a small but significant association with BVI (1.9%; 95% CI 1.1, 2.6; p<0.001).

Our ad hoc analysis of pwCF with at least one G551D mutation (N = 1,208) showed sufficient data to model BVI in this population, with a subgroup of 1,031 pwCF contributing data between 2004–2011 and 1,075 between 2012–2016 (Panel a in S2 Fig). As a note, due to the longitudinal nature of the data, many individuals contributed to both timeframes. We observed that pwCF with the G551D mutation had on average 5.8% longer intervals between visits after 2012 (Panel b in S2 Fig), although confidence intervals are wide and overlap across most of the lifespan.

## Discussion

To our knowledge, our analysis of BVI predictors at US CF centers is the largest and most comprehensive to date in this unique population. While average changes in BVI over the lifespan of pwCF may appear modest, we observed a substantial decrease in visit frequency between ages 18–25. During this critical time, 59% of pwCF had care gaps of ≥6 months, with 25% experiencing gaps of ≥1 year. These findings are particularly concerning given our prior research that showed prolonged gaps in care at this age have the largest negative effects on lung function [1]. Further investigation is needed to determine optimal visit frequency during this critical period, a time when many are facing changing life circumstances and may be difficult to contact. The question of "how long is too long" during this pivotal period of development therefore warrants detailed investigation.

BVI varied widely by insurance type. While publicly and privately insured children attended within recommended intervals, privately insured pwCF on average deviated from the guidelines in early adulthood. Uninsured adults had the longest BVI, averaging 4-months. Our results showing wide variance in BVI by insurance type differ somewhat from prior studies. Schechter et al. found that while those covered by Medicaid had significantly worse health outcomes, the annual number of CF center visits mirrored that of pwCF on other insurance plans [14]. In contrast, we observed significant though small differences in BVI between public and private insurance programs, but these differences did not develop until after age 20, beyond the age ceiling examined by Schechter et al. Li et al. found that adult pwCF with public insurance were more likely than those with private plans to meet the CFF yearly recommendations for clinic visits, respiratory cultures and PFT frequency [15]. The use of a composite outcome, exclusion of pwCF with milder lung disease, and other methodological differences make it difficult to directly compare the two studies. However, our finding that patients with more severe disease attend clinic more frequently is broadly consistent with their results.

Rurality only weakly associated with increased BVI in our analysis, consistent with prior analysis of USCFFPR data [14]. Still, this is surprising, given the patterns seen more generally in the US healthcare system [16]. Commuting time to care center, rather than the rurality of residence, may correlate better with BVI and is worthy of future investigation.

As has been found in other chronic conditions [17–19], our analysis shows that males with CF, particularly in early adulthood, attended clinic visits less frequently. This may reflect known sex differences in CF outcomes [20–23]. The convergence of BVI curves post-menopause merits further evaluation.

PwCF without complications had longer BVI, raising questions about patient preferences, physician recommendations, and barriers to care. Adults with CF face high treatment burdens [24] and financial hardships [25], which may impact visit frequency [25, 26]. While prolonged gaps in care negatively affect lung function [1, 3–5], it is unclear if increasing visit frequency will substantially improve outcomes or be desirable to patients. A quality improvement project at Arkansas Children's Hospital resulted in a remarkable increase in four-visit-per-year clinic attendance from 35% to 90%. The intervention was described as "labour intensive", requiring a redesign of the scheduling system, systematic attendance monitoring, and patient/family education. This yielded a small improvement in BMI percentile; however, other outcomes such as lung function and patient satisfaction were not measured [5]. More evidence is needed to fine tune visit frequency, and optimal BVI likely will vary by patient profile and across the lifespan.

Study limitations include reliance on USCFFPR data quality, with potential underreporting of visits and broad categorization of sociodemographic variables. The USCFFPR is reported to capture 95% of clinic visits and highly accurate data about key variables of interest, such as lung function and nutritional status [7]; however, the quality of other variables used in this study have not been evaluated. Since we wanted to understand the effect of education on BVI across the lifespan, we combined educational attainment of individual family members into one variable (highest education obtained by patient, spouse, and parents at each encounter); however, this approach may have introduced noise, possibly contributing to the weak signal observed. More detailed and complete data could reveal additional BVI patterns. We imputed missing data using LOCF/NOCB, a method well-suited to the complex structure of our longitudinal, hierarchical data. This approach, recommended by USCFFPR guidelines [27], effectively handled the time-dependent nature of our observations while maintaining computational efficiency. The majority of variables in our study had a low proportion of missing data, requiring minimal imputation. We have provided details on missing data frequencies and imputation rates in S1 Table. Additionally, we conducted a complete case analysis, which excluded a substantial portion of the data but yielded similar results for disease-related characteristics, reinforcing the robustness of our findings. Log-transformation of BVI mitigated potential issues with extreme values, resulting in an approximately normal distribution. Our large cohort yielded significant *P values* even for small effects, and we emphasize the importance of interpreting our findings based on effect size and clinical relevance, not just statistical significance.

We acknowledge that we are looking at visit frequency in this study and do not address visit quality, visit length, use of recommended treatments, or patient experience of care. Increased visit frequency does not necessarily correlate with improved outcomes unless care is effectively tailored to the patient's needs. These are other important modifiers of patient outcomes that are worthy of further investigation although some would require integration of data outside the existing USCFFPR.

Our study predates elexacaftor/tezacaftor/ivacaftor (ETI) approval and the COVID-19 pandemic, both of which changed the landscape of CF care. ETI has improved outcomes and reduced treatment burden [28], prompting experts to reconsider existing visit frequency recommendations [29, 30]. We expect that pwCF experiencing the clinical benefits of highly-effective modulators like ETI are electing for less frequent visits, which would result in increased BVI. This is supported by our exploratory analysis of individuals with the G551D mutation, which showed a small but noteworthy increase of 6% in BVI in this subset following FDA approval of ivacaftor in 2012. Oates and Schechter predicted that as the median age of survival for pwCF continues to rise, socioeconomic disparities in CF care will become more pronounced, emphasizing the importance of identifying and addressing the social determinants

that significantly impact CF outcomes [31]. Additionally, the COVID-19 pandemic prompted widespread adoption of telehealth [27, 32], potentially reducing access barriers for some while exacerbating disparities for others. Although our study was conducted before these changes in CF care, its findings provide a valuable baseline for evaluating shifts in healthcare utilization patterns. This historical context allows for a more nuanced assessment of how new therapies and care delivery models are reshaping CF management and patient outcomes.

Future research should evaluate predictors of care utilization in larger ETI-treated populations, examine factors influencing telehealth versus in-person care utilization, and assess the long-term impact of evolving care models on clinical outcomes and healthcare utilization. These studies will be crucial for optimizing CF care delivery in the era of highly-effective modulators.

Understanding BVI predictors can help streamline existing protocols for healthy pwCF, while supplying evidence to intensify plans for those at high risk of missing consequential elements of specialty care. By leveraging 13 years of USCFFPR data from over 28,000 pwCF and 800,000 separate encounters, we offer unique insights into the complex interplay of sociodemographic, clinical, and temporal factors influencing CF care utilization. This research not only fills a crucial knowledge gap but also provides a foundation for evidence-based interventions to enhance CF care delivery and patient engagement.

## Supporting information

**S1 Fig. A directed acyclic graph (DAG) that illustrates the hypothesized causal relationships among the covariates included in our statistical models.** The DAG guided the selection and adjustment of variables in our analyses to account for potential confounding pathways. This version of the DAG is color coded to represent one of our models, the analysis of the relationship between underweight BMI and between visit interval (BVI). White circles are variables that comprise the minimally sufficient adjustment set and are adjusted for in our model. Red denotes an ancestor of exposure *and* outcome, while blue denotes an ancestor of outcome only. Black lines denote causal relationships between variables, and green lines represent the causal relationship of interest (direct and indirect pathways). Figure created using DAGitty (http://dagitty.net/).
(PDF)

**S2 Fig. Findings from an ad hoc analysis evaluating changes in BVI in those with at least one G551D mutation before and after 2012 FDA approval of ivacaftor.** Panel a.) shows the number of individuals with the G551D mutation by age, before (light green) and after 2012 (dark green), b.) shows estimated between-visit interval across the lifespan before and after 2012 FDA approval of ivacaftor. Color shaded areas indicate 95% confidence bounds. The gray shaded area indicates the current CF Foundation-recommended between-visit interval of 90 days for reference.
(PDF)

**S1 Table. Study variable definitions.**
(PDF)

**S2 Table. Characteristics of people with cystic fibrosis age 6–60 years by inclusion status.** Individual visits were excluded if pulmonary function testing (PFT) results or body mass index (BMI) was missing. Individuals were excluded if they had fewer than 2 visits between 2004–2016.
(PDF)

**S3 Table. Length of between visit interval across study period.**
(PDF)

**S4 Table. Frequency of prolonged gaps by age group.**
(PDF)

**S5 Table. Frequency of prolonged gaps by pediatric age brackets.**
(PDF)

**S6 Table. Confounding adjustment.** Confounding adjustment sets for the association between each predictor of interest and the study outcome, between visit interval.
(PDF)

**S7 Table. Sensitivity analyses.** Sensitivity analyses evaluating multivariable regression results from the complete case analysis, after removing the between visit interval < 30 days exclusion criterion, and in a subset of people who are not experiencing CF-related complications (severe lung impairment defined as FEV1PP $\leq$ 40, underweight BMI, CF-related diabetes, and chronic infections).
(PDF)

**S8 Table. Sensitivity analysis evaluating multivariable results stratified by age group.** Pediatric is defined as <18 and adults are $\geq$18 years of age.
(PDF)

**S1 File. Model specifications, example code, and description of sensitivity analyses.**
(PDF)

## Acknowledgments

The authors thank the Cystic Fibrosis Foundation for the use of United States CF Foundation Patient Registry (USCFFPR) data to conduct this study. Additionally, the authors thank the people with CF, care providers, and clinic coordinators at CF centers throughout the USA for their contributions to the USCFFPR.

## Author Contributions

**Conceptualization:** Alexandra C. Hinton, Jonathan B. Zuckerman.

**Data curation:** Alexandra C. Hinton.

**Formal analysis:** Alexandra C. Hinton.

**Funding acquisition:** Alexandra C. Hinton, Jonathan B. Zuckerman.

**Methodology:** Alexandra C. Hinton, Edmund H. Sears, Jonathan B. Zuckerman, Sara Lopez-Pintado.

**Supervision:** Edmund H. Sears, Jonathan B. Zuckerman, Sara Lopez-Pintado.

**Visualization:** Alexandra C. Hinton.

**Writing – original draft:** Alexandra C. Hinton.

**Writing – review & editing:** Alexandra C. Hinton, Edmund H. Sears, Jonathan B. Zuckerman, Sara Lopez-Pintado.

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
