## [Decision Letter · Decision Letter 0]

11 Oct 2024

PONE-D-24-33750PREDICTORS OF FREQUENCY OF CF CARE IN THE US CYSTIC FIBROSIS FOUNDATION PATIENT REGISTRYPLOS ONE

Dear Dr. Hinton,

Thank you for submitting your manuscript to PLOS ONE. After careful consideration, we feel that it has merit but does not fully meet PLOS ONE’s publication criteria as it currently stands. Therefore, we invite you to submit a revised version of the manuscript that addresses the points raised during the review process.

We look forward to receiving your revised manuscript.

Kind regards,

Abdelwahab Omri, Pharm B, Ph.D, Laurentian University

Academic Editor

PLOS ONE

Journal Requirements:

“The authors thank the Cystic Fibrosis Foundation for the use of United States CF Foundation Patient Registry (USCFFPR) data to conduct this study and for and StatNet grant 003847Y7122. Additionally, the authors thank the people with CF, care providers, and clinic coordinators at CF centers throughout the USA for their contributions to the USCFFPR. Data are available upon request through the USCFFPR Comparative Effectiveness Research Committee. The committee can be contacted at datarequests@cff.org. “

“AM and JZ received funding from the Cystic Fibrosis Foundation (https://www.cff.org/) through the StatNet grant 003847Y7122.  Funders did not play any role in the study design, data collection and analysis, decision to publish, or preparation of the manuscript.”

 3. We noted in your submission details that a portion of your manuscript may have been presented or published elsewhere. [Preliminary results were published as an abstract associated with the 2023 North American Cystic Fibrosis Conference. A. Hinton ES, S. Lopez-Pintado, J. Zuckerman. 614 Time between clinic visits of individuals with cystic fibrosis: patterns across the lifespan. Journal of Cystic Fibrosis. 2023;22:S327-S328. doi:https://doi.org/10.1016/S1569-1993(23)01536-9

Results, data, or figures are not published elsewhere.] Please clarify whether this [conference proceeding or publication] was peer-reviewed and formally published. If this work was previously peer-reviewed and published, in the cover letter please provide the reason that this work does not constitute dual publication and should be included in the current manuscript.

4. In the online submission form, you indicated that [Data are available upon request through the USCFFPR Comparative Effectiveness Research Committee. The committee can be contacted at datarequests@cff.org.].

Reviewers' comments:

Reviewer's Responses to Questions

**Comments to the Author**

1. Is the manuscript technically sound, and do the data support the conclusions?

Reviewer #1: Yes

Reviewer #2: Yes

2. Has the statistical analysis been performed appropriately and rigorously? 

Reviewer #1: Yes

Reviewer #2: Yes

3. Have the authors made all data underlying the findings in their manuscript fully available?

Reviewer #1: Yes

Reviewer #2: Yes

4. Is the manuscript presented in an intelligible fashion and written in standard English?

Reviewer #1: Yes

Reviewer #2: Yes

5. Review Comments to the Author

Reviewer #1: Thank you for your very interesting study that I careful read with great pleasure.

Your observational cohort study, conducted using data of the USCFF patients’ registry, aimed to investigate sociodemographic and disease-related factors predictive of visit frequency in people with CF and to assess how these effects vary across the lifespan. The relationship between patient-level characteristics and between-visit interval (BVI) was modeled using multivariable longitudinal semiparametric regression.

The study, with an important data collection on more than 28,000 subjects with more than 850,000 encountered, showed that 55% of visits (in pwCF 6-60 years old) occurred within 90 days of the prior visit, adhering to national guidelines. Males, non-white individuals, uninsured, especially young adults, and subjects without CF complications had a longer BVI, while close visits in patients with complications reflect their greater needs for care.

The study also identifies categories of patients with intervals between visits greater than 6 or 12 months (59% and 25% respectively in the 18-25 age group), intervals that may increase the risk of greater lung damage.

I found the study, and your paper, complex but comprehensive, and it provides a lot of information to help clinicians undertake quality improvement programs for non-standardized BVI, thus CF patient management could be designed on characteristics and needs of patients, improving CF care delivery and somehow an increase in patients' awareness of their needs.

There are two typo, line 114, pwCF � CF and line 124 QI � quality improvement (QI)

Reviewer #2: This paper, authored by an experienced group, focuses on the frequency of patient visits to cystic fibrosis (CF) centers, drawing from a large cohort in the US Cystic Fibrosis Foundation Patient Registry (USCFFPR). The study offers valuable insights and makes a strong case for refining guidelines to promote more frequent and tailored care, particularly for those at risk of poor outcomes due to sociodemographic and insurance-related factors. Its findings are particularly relevant for CF clinicians, caregivers, health insurers, and researchers in healthcare delivery.

However, there are a few key points that the authors should address before publication:

ETI's Impact on CF Care: The data was collected before the introduction of elexacaftor/tezacaftor/ivacaftor (ETI), which has revolutionized CF treatment. This should be discussed in more detail, as it could significantly impact patient care moving forward.

In line with the above mentioned point: Ivacaftor and Visit Frequency: Ivacaftor was approved in 2012 for patients with at least one G551D mutation, which may have influenced visit patterns. Can you extrapolate between-visit intervals (BVI) for this specific group and compare visit frequency before and after the introduction of Ivacaftor? Addressing whether this therapy impacted BVI would enhance the relevance of your findings.

Quality vs. Frequency of Visits: While the study emphasizes visit frequency, it lacks an assessment of the quality of care provided during those visits. Increased visit frequency does not always correlate with improved outcomes unless care is effectively tailored to the patient’s needs. It would be valuable to include data on how many patients received the standard of care regarding medications. Additionally, could patients who are unable to afford necessary drugs be skipping visits due to frustration over diagnostic results that do not lead to receiving appropriate therapy? Discussing this would add depth to the analysis.

Patient-Reported Outcomes: The abstract does not mention patient-reported outcomes, such as satisfaction with care or perceived barriers to attending visits. Including patient perspectives would provide important context for understanding the observed disparities. Please consider addressing this in the discussion section.

Potential Data Limitations: There is a possibility that visit frequency was underreported, and some sociodemographic variables may have been categorized too broadly. Clarifying the extent of this issue could help address concerns about data completeness and accuracy.

Data Imputation Methods: The use of Last Observation Carried Forward (LOCF) and Next Observation Carried Backward (NOCB) for imputing missing data may introduce bias. Please discuss how these methods might have influenced your findings.

By addressing these points, the paper would offer a more comprehensive understanding of the factors influencing care frequency and the impact of recent treatment advancements on CF management.

6. PLOS authors have the option to publish the peer review history of their article (what does this mean?). If published, this will include your full peer review and any attached files.

Reviewer #1: No

Reviewer #2: **Yes: **Dr. Olaf Eickmeier

---

## [Author Response · Author response to Decision Letter 0]

22 Oct 2024

Dear Reviewers,

Thank you for the opportunity to revise and resubmit our manuscript titled "PREDICTORS OF FREQUENCY OF CF CARE IN THE US CYSTIC FIBROSIS FOUNDATION PATIENT REGISTRY" (Manuscript ID: PONE-D-24-33750) for consideration in PLOS ONE. We appreciate your constructive feedback which has helped us improve the quality of our paper. 

We have carefully addressed comments and suggestions you provided. A detailed point-by-point response can be found below. We believe these revisions have significantly strengthened our paper and hope that it now meets the standards for publication in PLOS ONE. We look forward to your decision and would be happy to address further questions or concerns. 

Reviewer #1: Thank you for your very interesting study that I careful read with great pleasure.

Your observational cohort study, conducted using data of the USCFF patients’ registry, aimed to investigate sociodemographic and disease-related factors predictive of visit frequency in people with CF and to assess how these effects vary across the lifespan. The relationship between patient-level characteristics and between-visit interval (BVI) was modeled using multivariable longitudinal semiparametric regression.

The study, with an important data collection on more than 28,000 subjects with more than 850,000 encountered, showed that 55% of visits (in pwCF 6-60 years old) occurred within 90 days of the prior visit, adhering to national guidelines. Males, non-white individuals, uninsured, especially young adults, and subjects without CF complications had a longer BVI, while close visits in patients with complications reflect their greater needs for care.

The study also identifies categories of patients with intervals between visits greater than 6 or 12 months (59% and 25% respectively in the 18-25 age group), intervals that may increase the risk of greater lung damage.

I found the study, and your paper, complex but comprehensive, and it provides a lot of information to help clinicians undertake quality improvement programs for non-standardized BVI, thus CF patient management could be designed on characteristics and needs of patients, improving CF care delivery and somehow an increase in patients' awareness of their needs.

There are two typo, line 114, pwCF à CF and line 124 QI à quality improvement (QI)

Response: Thank you for your careful reading and positive feedback on the comprehensiveness and clinical relevance of our work. We appreciate you pointing out the two typos. We have corrected "people with pwCF" to "pwCF" which now appears on line 52 in the revised manuscript, and we spell out "quality improvement" on line 62 in the revised version. 

Reviewer #2: This paper, authored by an experienced group, focuses on the frequency of patient visits to cystic fibrosis (CF) centers, drawing from a large cohort in the US Cystic Fibrosis Foundation Patient Registry (USCFFPR). The study offers valuable insights and makes a strong case for refining guidelines to promote more frequent and tailored care, particularly for those at risk of poor outcomes due to sociodemographic and insurance-related factors. Its findings are particularly relevant for CF clinicians, caregivers, health insurers, and researchers in healthcare delivery.

However, there are a few key points that the authors should address before publication:

Ø ETI's Impact on CF Care: The data was collected before the introduction of elexacaftor/tezacaftor/ivacaftor (ETI), which has revolutionized CF treatment. This should be discussed in more detail, as it could significantly impact patient care moving forward.

Response: This is an important point, and we appreciate the opportunity to bolster our discussion of this in the manuscript. We have added the following to the discussion section (Lines 395-409):

“Our study predates elexacaftor/tezacaftor/ivacaftor (ETI) approval and the COVID-19 pandemic, both of which changed the landscape of CF care. ETI has improved outcomes and reduced treatment burden. We expect that pwCF experiencing the clinical benefits of highly-effective modulators like ETI are electing for less frequent visits, which would result in increased BVI. This is supported by our exploratory analysis of individuals with the G551D mutation, which showed a small but noteworthy increase of 6% in BVI in this subset following FDA approval of ivacaftor in 2012. Oates and Schechter predicted that as the median age of survival for pwCF continues to rise, socioeconomic disparities in CF care will become more pronounced, emphasizing the importance of identifying and addressing the social determinants that significantly impact CF outcomes. Additionally, the COVID-19 pandemic prompted widespread adoption of telehealth, potentially reducing access barriers for some while exacerbating disparities for others. Although our study was conducted before these changes in CF care, its findings provide a valuable baseline for evaluating shifts in healthcare utilization patterns. This historical context allows for a more nuanced assessment of how new therapies and care delivery models are reshaping CF management and patient outcomes.”

Ø In line with the above mentioned point: Ivacaftor and Visit Frequency: Ivacaftor was approved in 2012 for patients with at least one G551D mutation, which may have influenced visit patterns. Can you extrapolate between-visit intervals (BVI) for this specific group and compare visit frequency before and after the introduction of Ivacaftor? Addressing whether this therapy impacted BVI would enhance the relevance of your findings.

Response: Thank you for this helpful suggestion. While we do not have data on ivacaftor use, we do have mutation information, and uptake of ivacaftor for patients with G551D was rapid.(1) Based on your feedback, we have analyzed differences in BVI in people with at least one G551D mutation before (2004-2011) and after (2012-2016) FDA-approval of ivacaftor. We have included our methods (Lines 154-161) and a brief description of our findings in the manuscript (Lines 315-320) and included figures in the supplement (Panel a & b in S2 Fig). 

In brief, we found a small but significant increase in BVI post ivacaftor approval in people with at least one G551D mutation. This provides some evidence of trends that may emerge with triple combination modulator therapy (e.g. elexacaftor/tezacaftor/ivacaftor), which was FDA-approved following the study window of this investigation. While we did not evaluate predictors of BVI post-2012 in pwCF taking CFTR modulators, we note in the discussion that future research should explore this.

Ø Quality vs. Frequency of Visits: While the study emphasizes visit frequency, it lacks an assessment of the quality of care provided during those visits. Increased visit frequency does not always correlate with improved outcomes unless care is effectively tailored to the patient’s needs. It would be valuable to include data on how many patients received the standard of care regarding medications. Additionally, could patients who are unable to afford necessary drugs be skipping visits due to frustration over diagnostic results that do not lead to receiving appropriate therapy? Discussing this would add depth to the analysis.

Response: Thank you for your insightful feedback on our study. We acknowledge the importance of assessing the quality care in addition to the frequency of visits in CF care. Our focus on visit frequency was intended as an initial step in understanding care utilization patterns, and we appreciate the importance of factors such as visit quality, visit length, use of recommended treatments, or patient experience of care are crucial for improving patient outcomes. You also raise valid concerns about the potential impact of treatment affordability on visit adherence. While the current study does not directly address these aspects, we agree that incorporating such information would provide a more comprehensive picture of CF care delivery and warrant further investigation. We have added a discussion about this and your comment below regarding patient-reported outcomes to our manuscript (Lines 389-393).

Ø Patient-Reported Outcomes: The abstract does not mention patient-reported outcomes, such as satisfaction with care or perceived barriers to attending visits. Including patient perspectives would provide important context for understanding the observed disparities. Please consider addressing this in the discussion section.

Response: You raise important points regarding patient experience of care which are crucial for improving patient outcomes. Patient perspectives are not directly collected in the USCFFPR and would also require substantial additional data collection and analysis. We plan to explore in future research. As mentioned above, please see lines 389-393 for a discussion of these issues in the manuscript.

Ø Potential Data Limitations: There is a possibility that visit frequency was underreported, and some sociodemographic variables may have been categorized too broadly. Clarifying the extent of this issue could help address concerns about data completeness and accuracy.

Response: We have noted that studies of the USCFFPR accuracy have shown the registry contains 95% clinic visits and highly accurate data about key variables of interest, such as lung function and nutritional status (Knapp, et al., 2016); however, the quality of other variables used in this study has not been evaluated. Regarding the use of broad categorizations, we note in our manuscript (lines 375-376): “however, this approach may have introduced noise, possibly contributing to the weak signal observed.” Data inaccuracies or missing data are more likely to obscure true relationships rather than create false ones. This conservative bias typically results in underestimating the strength of associations rather than overestimating them. We have also addressed this reviewer’s concern by adding information into S1 Table regarding the frequency of missing data and the proportion that were imputed.

Ø Data Imputation Methods: The use of Last Observation Carried Forward (LOCF) and Next Observation Carried Backward (NOCB) for imputing missing data may introduce bias. Please discuss how these methods might have influenced your findings.

Response: We acknowledge that LOCF/NOCB methods may introduce bias, and we carefully considered the implications on our findings. In our study, these methods were used to address missing data in the following categories: BMI, FEV1PP, family income, education, and rurality. We have added data to S1 Table specifying the amount of missing data and the proportion that were imputed. For our primary analyses, we include records where imputation was not possible by creating and including missing or unknown categories in our model (See Tables 1 and 2). We believe the impact on our main conclusions regarding BMI, rurality, and education is limited due to the low proportion of missing data (5% of encounters had imputed BMI and 7% had imputed rurality, no actual effect of imputation on education). For lung impairment (measured by FEV1PP), 12% of encounters had imputed data. Lung impairment showed a strong and logical association with time between visits. When we ran the model with missing data coded as a separate category rather than imputing, the findings were very similar and qualitatively the same. The estimate for the missing category fell between those for moderate and severe lung impairment. Importantly, lung impairment was not used for confounding adjustment in any other models. Income was missing for 77% of encounters and was imputed for 37% (imputation could not be done when individuals missing data for a particular variable across the entire study period). Analyses excluding encounters missing income showed very similar results. Unfortunately, income was in the confounding set for analyses evaluating the effect of insurance, insurance and race, and underweight BMI on the between visit interval. However, findings were qualitatively similar for each of these when excluding income from the adjustment set. Additionally, we present a complete case analysis in the supplement (S7 Table). This analysis excludes all encounters where any one of our variables of interest were missing prior to imputation, which resulted in the exclusion of 81% of encounters and 48% of individuals. We saw similar results for disease-related characteristics but showed longer Between Visit Intervals (BVI) for older age, advanced education, and higher income in the complete case analysis. This discrepancy actually supports our decision to impute values, as the complete cases likely overrepresented individuals with higher socioeconomic status and better healthcare access, potentially biasing results.

For longitudinal clinical data, we expect a degree of consistency over time, in which more temporally proximate measures are more alike than those separated by more time. LOCF and NOCB methods maintain the time-dependent nature of observations, which is important for understanding how disease progression affects visit frequency over time. We considered multiple imputation as an alternative approach; however, analytic complexity would be greatly increased in the context of the time-varying, hierarchical structure of our USCFFPR dataset. Additionally, this approach was noted as appropriate for this registry by Dasenbrook and Sawicki.(2) While we acknowledge the potential limitations of LOCF/NOCB, we believe its use in this study was justified given the nature of our data and the research questions. We are transparent about our methods and discuss this topic in our limitations section.

1. Sawicki GS, Dasenbrook E, Fink AK, Schechter MS. Rate of Uptake of Ivacaftor Use after U.S. Food and Drug Administration Approval among Patients Enrolled in the U.S. Cystic Fibrosis Foundation Patient Registry. Ann Am Thorac Soc. 2015;12(8):1146-52.

2. Dasenbrook EC, Sawicki GS. Cystic fibrosis patient registries: A valuable source for clinical research. J Cyst Fibros. 2018;17(4):433-40.

---

## [Editor Report · Decision Letter 1]

25 Oct 2024

PREDICTORS OF FREQUENCY OF CF CARE IN THE US CYSTIC FIBROSIS FOUNDATION PATIENT REGISTRY

PONE-D-24-33750R1

Dear Dr. Alexandra C Hinton,

We’re pleased to inform you that your manuscript has been judged scientifically suitable for publication and will be formally accepted for publication once it meets all outstanding technical requirements.

Kind regards,

Abdelwahab Omri, Pharm B, Ph.D, Laurentian University

Academic Editor

PLOS ONE

---

## [Editor Report · Acceptance letter]

20 Nov 2024

PONE-D-24-33750R1 

PLOS ONE

Dear Dr. Hinton, 

I'm pleased to inform you that your manuscript has been deemed suitable for publication in PLOS ONE. Congratulations! Your manuscript is now being handed over to our production team.

Kind regards, 

on behalf of

Dr. Abdelwahab Omri 

Academic Editor

PLOS ONE